# Development and validation of a life expectancy estimator for multimorbid older adults: a cohort study protocol

Viktoria Gastens ![ORCID],[1,2] Cinzia Del Giovane,[1,3] Daniela Anker,[1,3] Martin Feller,[1,4] Lamprini Syrogiannouli,[1] Nathalie Schwab,[1,4] Douglas C Bauer,[5] Nicolas Rodondi,[1,4] Arnaud Chiolero[3,6]

[1]Institute of Primary Health Care (BIHAM), University of Bern, Bern, Switzerland
[2]Graduate School for Health Sciences, University of Bern, Bern, Switzerland
[3]Population Health Laboratory (#PopHealthLab), University of Fribourg, Fribourg, Switzerland
[4]Department of General Internal Medicine, Inselspital, Bern University Hospital, University of Bern, Bern, Switzerland
[5]Departments of Medicine and Epidemiology & Biostatistics, University of California San Francisco, San Francisco, California, USA
[6]School of Global and Population Health, McGill University, Montreal, Quebec, Canada

**Correspondence to**
Viktoria Gastens;
viktoria.gastens@biham.unibe.ch

## ABSTRACT

**Background** Older multimorbid adults have a high risk of mortality and a short life expectancy (LE). Providing high-value care and avoiding care overuse, including of preventive care, is a serious challenge among multimorbid patients. While guidelines recommend to tailor preventive care according to the estimated LE, there is no tool to estimate LE in this specific population. Our objective is therefore to develop an LE estimator for older multimorbid adults by transforming a mortality prognostic index, which will be developed and internally validated in a prospective cohort.

**Methods and analysis** We will analyse data of the Optimising Therapy to Prevent Avoidable Hospital Admissions in Multimorbid Older People cohort study in Bern, Switzerland. 822 participants were included at hospitalisation with age of 70 years or older, multimorbidity (three or more chronic medical conditions) and polypharmacy (use of five drugs or more for >30 days). All-cause mortality will be assessed during 3 years of follow-up. We will apply a flexible parametric survival model with backward stepwise selection to identify the mortality risk predictors. The model will be internally validated using bootstrapping techniques. We will derive a point-based risk score from the regression coefficients. We will transform the 3-year mortality prognostic index into an LE estimator using the Gompertz survival function. We will perform a qualitative assessment of the clinical usability of the LE estimator and its application. We will conduct the development and validation of the mortality prognostic index following the Prognosis Research Strategy (PROGRESS) framework and report it following the Transparent Reporting of a Multivariable Prediction Model for Individual Prognosis or Diagnosis (TRIPOD) statement.

**Ethics and dissemination** Written informed consent by patients themselves or, in the case of cognitive impairment, by a legal representative, was required before enrolment. The local ethics committee (Kantonale Ethikkommission Bern) has approved the study. We plan to publish the results in peer-reviewed journals and present them at national and international conferences.

### Strengths and limitations of this study

► We will provide the first life expectancy estimator specifically for older multimorbid adults.
► We use high-quality data from a large prospective cohort study of multimorbid older adults.
► This predictive tool may help clinicians to personalise preventive care according to individual life expectancy.
► External validation of the new life expectancy estimator in another cohort will still be required.

healthcare utilisation such as polypharmacy[2] and multiple hospital admissions.[3] While the benefits and harms of preventive care, such as cancer screening and cardiovascular disease (CVD) preventive treatment, are well known up to the age of 75 years, they are not directly applicable to older multimorbid patients.[4 5] For this reason, international guidelines such as US Preventive Services Task Force recommend accounting for LE when deciding on preventive care.

Accounting for LE is indeed key because most preventive care has a lag time to benefit. The lag time to benefit is the time interval between an intervention (eg, cancer screening) and the moment when health outcomes are improved (eg, lower cancer mortality). It varies greatly, ranging from 6 months for statin treatment in the secondary prevention of CVD to up to 10 years for several cancer screenings.[6 7] If the lag time to benefit of the intervention is longer than the estimated LE, the patient cannot benefit from it[8 9] and the intervention should not be recommended. Older multimorbid adults are at high risk of not having the time to benefit, and harms may outweigh the benefits of an intervention. To decide which preventive care should be delivered, the lag time to benefit should be therefore weighted against

## BACKGROUND

Multimorbidity is frequent among older adults and is associated with a shorter life expectancy (LE),[1] a low quality of life and a high

the estimated LE, as proposed for cancer screening by Lee *et al*.[6 7]

There is, however, no tool to accurately estimate LE among multimorbid patients. LE estimators are usually derived from mortality risk scores. Several risk scores allow estimating mortality risk for older adults,[8] but they are not recommended for widespread use and do not directly translate to LE estimates. To our knowledge, only Lee *et al* have developed an LE estimator from a mortality risk prognostic index from a cohort of community-dwelling older adults with a relatively low mortality rate,[9] but it is not directly applicable to multimorbid patients and other populations with relatively high mortality rates. Our objective is therefore to develop and internally validate an LE estimator for older multimorbid adults.

## METHODS AND ANALYSIS
### Source of data and study design
We will use data from 822 participants of an ongoing study in Bern, Switzerland. Participants were originally enrolled in a clinical trial with an intervention lasting 1 year and will be followed up postintervention for an additional 2 years as a cohort. The clinical trial was the cluster randomised controlled trial Optimising Therapy to Prevent Avoidable Hospital Admissions in Multimorbid Older People (OPERAM[10]). Participants were assigned to receive either standard care or a medication review by a Systematic Tool to Reduce Inappropriate Prescribing with observation of the primary outcome of drug-related hospital admission over 1 year. For our study, we will use data collected at baseline (December 2016 to October 2018) and up to 3 years after baseline (until October 2021). Figure 1 shows the schedule of data collection for this project.

Study nurses collected baseline data by a personal interview with the participant and from medical files. Follow-ups are conducted via phone calls. Phone interviews are held with participants or relatives, otherwise with a proxy or with the general practitioner, when the participants are not reachable or not able to answer.

### Participants
Eight hundred and twenty-two participants were enrolled at hospitalisation in the Inselspital, University Hospital, Bern, Switzerland. Inclusion criteria were age 70 years or older, multimorbidity (three or more chronic medical conditions) and polypharmacy (use of five drugs or more for >30 days). We defined multimorbidity as presence of three or more coexistent chronic medical conditions with an estimated duration of 6 months or more. The medical conditions were recorded as International Classification of Diseases (ICD-10) codes. The estimated duration was based on a clinical decision. The 10 most frequent chronic medical conditions were in descending order: essential hypertension (I10), type 2 diabetes mellitus (E11), hypertensive heart disease (I11), chronic kidney disease stage 3 (N18.3), disorders of lipoprotein metabolism and other lipidaemias (E78), atrial fibrillation and flutter (I48), chronic ischaemic heart disease (I25), hyperplasia of prostate (N40), other postsurgical states (Z98) and obesity (E66). A detailed account of the study participants' baseline characteristics is given in table 1.

Written informed consent by patients themselves or, in the case of cognitive impairment, by a legal representative, had already been obtained before enrolment. Patients planned for direct admission to palliative care (<24 hours after admission), or patients undergoing a structured drug review other than the trial intervention, or who had passed a structured drug review within the last 2 months were excluded. Also patients for whom it was not possible to obtain an informed consent were excluded.

| Inclusion:<br>• ≥70 years<br>• Multimorbidity (≥3 conditions)<br>• Polypharmacy (≥5 drugs) | | Baseline<br><br>First patient first visit 12/2016 | 1 year follow-up | 2 year follow-up | 3 year follow-up<br>Last patient last visit 10/2021 |
|---|---|---|---|---|---|
| | Baseline characteristics | X | | | |
| Exclusion:<br>• Admission to palliative care (<24 h)<br>• Structured drug review (<2 months)<br>• No informed consent | Activities of Daily Living (ADL) | X | X | | |
| | Quality of Life (EQ-5D) | X | X | | |
| | Falls and fractures | X | X | | |
| | Health economics | X | X | | |
| | Diagnoses | X | X | X | X |
| | Medication | X | X | X | X |
| | Blood pressure and cholesterol | X | X | X | X |
| | Life expectancy questionnaire by Lee et al.[1] | | X | X | X |
| | Cancer screening | | X | X | X |

**Figure 1** Schedule of data collection from baseline to 3-year follow-up. [1]Lee *et al*.[15]

| Baseline characteristic | n (%) |
|---|---|
| **Table 1** Baseline characteristics of participants | |
| n | 822 |
| Female | 347 (42.2) |
| Age, median [IQR] | 79 [74.0; 84.0] |
| BMI, median [IQR] | 26.0 [23.0; 29.1] |
| Smoking | 69 (8.4) |
| Education | |
| University | 151 (18.4) |
| High school | 491 (59.7) |
| Less than high school | 170 (20.7) |
| Dementia | 82 (10.0) |
| History of cancer | 324 (39.4) |
| History of CVD | 528 (64.2) |
| CVD risk factors and medication | |
| Hypertension | 626 (76.2) |
| Hypertension with treatment | 529 (64.4) |
| Hypercholesterolaemia | 309 (37.6) |
| Hypercholesterolaemia with treatment | 229 (27.9) |
| Diabetes | 256 (31.1) |
| Diabetes with treatment | 202 (24.6) |
| Chronic medication, median [IQR] | 10 [6.5; 13.5] |
| Diagnoses, median [IQR] | 16 [12.5; 19.5] |
| Hospitalisations during the last year, median [IQR] | 1 [0.5; 1.5] |

BMI, body mass index; CVD, cardiovascular disease.

## Outcome

The outcome of interest is all-cause mortality over the 3 years of follow-up. One hundred and fifty-three deaths (18.6%) occurred over the first year of follow-up. Since participants were recruited during hospital admission, a higher mortality rate is expected during the first year of follow-up compared with subsequent follow-up and we expect approximately 250 deaths (30%) over the 3 years of follow-up.

## Development, validation and clinical usability of an LE estimator for multimorbid adults

We will develop and validate the mortality prognostic index following the Prognosis Research Strategy (PROGRESS) framework,[11] and report it following the Transparent Reporting of a Multivariable Prediction Model for Individual Prognosis or Diagnosis (TRIPOD) statement.[12] We will further follow the recommendations of Moons *et al*[13][14] for risk prediction models and apply the method described below for 3-year mortality prediction models. Figure 2 summarises the process of developing the LE estimator tool.

### Candidate predictors identification

Candidate predictors are usually derived from previous research efforts in this field,[8][15] ease and reliability of measurement in clinical setting and background knowledge on potential associations with the outcome. Therefore, we will perform a literature review of potential predictors, and also consider factors included in the OPERAM data set that may not be identified from the review but specific to the target population. Table 2 shows a list of potential candidate predictors. They include demographic (age, sex), clinical characteristics (comorbidity, medication, body mass index), smoking, functional status variables and hospitalisation. Notably, we will account for the severity of comorbidity and functional impairment, which has not been considered by Lee *et al*.[9]

As part of the OPERAM trial, the participants have been assigned to a control or intervention group at baseline. We will test the predictive effect of this assignment on mortality and account for it in the analyses if necessary.

### Mortality risk prognostic model

The relationship between candidate predictors and outcome will be analysed using the flexible parametric survival model. We will use the backward selection method.[13][14] The remaining variables will form the final model. We will use multiple imputation for missing values under a missing at random assumption in order to reduce bias and avoid excluding participants from the analysis.[13] We will investigate the predictive accuracy of

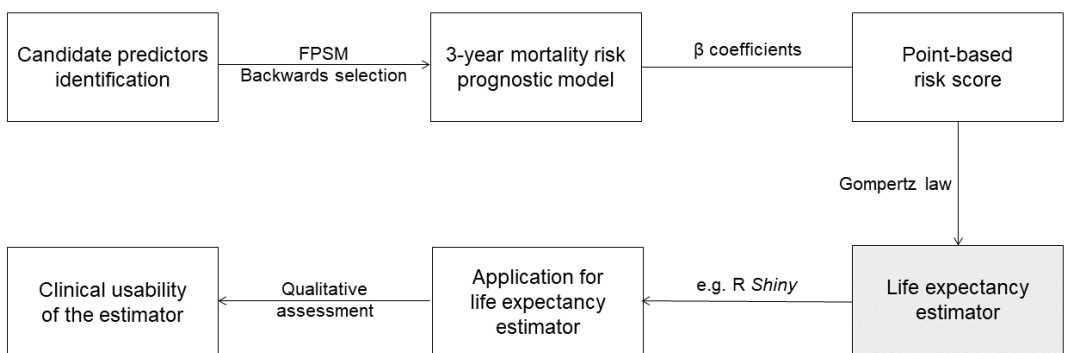

**Figure 2** Flow chart of the development steps (boxes) and methods used (arrows) for the life expectancy estimator tool. FPSM, flexible parametric survival model.

**Table 2** Potential candidate predictors from information obtained at baseline

| Variable | Description |
|---|---|
| Age | |
| Sex | |
| Comorbidities | Charlson Comorbidity Index (CCI) |
| Medication burden | Number of drugs before index hospitalisation |
| Body mass index (BMI) | Ratio of weight (kg)/square of height (m²) |
| Weight loss | Lost weight during last year (yes/no) |
| Smoking | Current smoker (yes/no) |
| Hospitalisation | Number of hospitalisations during last year |
| Activities of daily living (ADL) | Barthel Index: bathing, bladder control, bowel control, dressing, feeding, grooming, mobility, stairs, toilet use, transfer |
| Generic health status | EQ-5D score: mobility, self-care, usual activities, pain/discomfort and anxiety/depression |
| Falls | Number of falls during last year |
| Nursing | Nursing home residency (yes/no) |

**Table 3** Power to detect a statistically significant between-group difference in LE along various assumptions (n=274 per group; alpha=0.05; two sided)

| Difference in LE (years) | SD of LE (years) | Power (%) |
|---|---|---|
| 1 | 4 | 83 |
| 1 | 8 | 31 |
| 2 | 4 | >99 |
| 2 | 8 | 83 |
| 3 | 4 | >99 |
| 3 | 8 | >99 |

The relatively high power in most scenarios indicates that the sample size is adequate to address the research question. LE, life expectancy.

the final model by testing calibration and discrimination. We will assess the model's calibration by graphical calibration plots and by comparing the predicted to the observed mortality, with a relative difference of less than 10% considered satisfactory.[8] The model's discrimination will be assessed with Harrell's C statistic.[8] We will use bootstrapping techniques to internally validate our mortality prognostic index. We will perform 500 bootstrap cycles from the original sample, sampling the same number of patients with replacement. In each bootstrap sample, we will derive a mortality prediction model and the relative risk score, as done in the original sample. We will also evaluate potential overfitting and optimism. We will calculate optimism as difference in performance measure (eg, C statistics) between the original sample and the respective bootstrap sample. This will be repeated for all bootstrap samples to estimate the average optimism.[13 14] If necessary, we will adjust the model for overfitting.

We have calculated the required sample size for conducting our multivariable prediction model using the criteria proposed by Riley et al[16] and implemented in the *pmsampsize* library for the R environment.[17] For the 3-year mortality risk prognostic model, the minimum sample size required with 12 candidate predictor parameters, an expected outcome event rate of 0.1 per year and an anticipated Cox-Snell $R^2$ of 0.126 (C statistic of[15] 0.82[15 18]) is 799 with 20 events per predictor parameter. This is considerably more than the idiomatic 10 events per predictor parameter. Our sample size of 822 is therefore adequate for this project.

To have a better clinical sense of the statistical power of our study, we computed the power to detect a between-group difference in estimated LE that can be considered as important on a clinical point of view. If we stratify our sample in three groups of equal number of patients (n=274) with a relatively short, average and long estimated LE, the power to detect a statistically significant between-group difference of LE of 1–3 years is indicated in table 3.

### Point-based risk score

From the final model, we will derive a point-based risk score by assigning points to each risk factor. Each $\beta$ coefficient will be divided by the lowest $\beta$ coefficient and rounded to the nearest integer. The sum of points for each risk factor will then represent the total risk score of this participant.[15]

### LE estimator

We will transform the 3-year prognostic index into an LE estimator following the method of Lee et al.[9] In particular, we will use the new 3-year mortality prognostic index to define subpopulations with the same risk score.[9] We will fit a Gompertz survival function with each point score as a categorical predictor having a flexible proportional effect on the hazard rate. The Gompertz function assumes that each subpopulation will experience an exponential rise in mortality risk over time ($h_i(t)=\lambda_i \exp(\gamma t)$, where $\lambda_i \exp(x\beta)$). The Gompertz model will allow us to determine the time to 25%, 50% and 75% mortality for each of the risk point groups. We will report 95% confidence for the median survival intervals using bootstrapping techniques as well as 50% prediction intervals. We will compare our fit Gompertz model with observed Kaplan-Meier survival curves. The Lee LE estimator[9] will be recalibrated using standard methods to reflect the higher mortality rates of the target population.[14] The performance of the new LE estimator will be compared with the recalibrated Lee LE estimator by the Brier Score[19] and Harrell's C statistic. We will build an interactive web application of the final model using the Shiny package in the R environment.

## Clinical usability of the new LE estimator

We will perform a qualitative assessment of the LE estimator with one-to-one interviews with at least 10 clinicians.[20] We will assess whether the prediction parameters are pertinent and measurable in clinical practice, and evaluate barriers and enablers for impactful implementation of the tool. If no consent on the parameters to be included is found, we will use an iterative process based on the Delphi method.[21] Identified barriers and enablers will be used for the elaboration of implementation strategies.

## Patient and public involvement

Patients or the public were not involved in the design, or conduct, or reporting, or dissemination plans of our research.

## ETHICS AND DISSEMINATION

The local ethics committee (Kantonale Ethikkommission Bern) has approved the study protocol. All participants and their data are handled according to the ethical principles of the Declaration of Helsinki. This study complies with all applicable standards of the International Conference on Harmonisation E6 Guideline for Good Clinical Practice (1996) guideline. We plan to publish the results from our study in peer-reviewed and open access journals and present them at national and international conferences. Data will be deposited in the Bern Open Repository and Information System (www. boris.unibe.ch).

## PERSPECTIVE

To our knowledge, we will be the first to develop an LE estimator specifically for older multimorbid adults. A major strength of this study is its high feasibility due to the follow-up of 822 already included multimorbid participants with strong strategies of containment of loss to follow-up.[22] Notably, individuals with cognitive impairment were included, who are commonly excluded from studies. In addition to the development and internal validation of the LE estimator, we also assess its implementation and clinical usability.

One major limitation is the lack of external validation but we will explore opportunities to test the LE estimator in a different data set of older multimorbid adults. As the study participants were included at the time of hospitalisation, they may not be representative of all patients with multimorbidity, and this could reduce the transportability of our findings to other populations. Further, we have a relatively short duration of follow-up.

Our results will be useful for both clinical and research activities as they can have a major impact on preventive care practice by helping healthcare providers to tailor preventive care according to the estimated LE. Eventually, our study will help preventing underuse and overuse of preventive care in the growing older population.

**Contributors** Study concept and design: AC, NR, CDG. Acquisition, analysis, interpretation of data: NS, VG, CDG, LS. Drafting of the article: VG, CDG, AC. Critical revision of the article: VG, CDG, DA, MF, LS, NS, DCB, NR, AC. Statistical analysis: VG, LS, CDG. Obtained funding: AC. Administrative, technical or material support: NS. Supervision: AC, CDG.

**Funding** This work was supported by the Swiss National Science Foundation (grant number 320030_188549/01 to AC). This work is part of the project 'OPERAM: OPtimising thERapy to prevent Avoidable hospital admissions in the Multimorbid elderly' supported by the European Union's Horizon 2020 research and innovation programme (grant agreement number 6342388), and by the Swiss State Secretariat for Education, Research and Innovation (SERI) (contract number 15.0137).

**Disclaimer** The opinions expressed and arguments employed herein are those of the authors and do not necessarily reflect the official views of the EC and the Swiss government.

**Competing interests** None declared.

**Patient and public involvement** Patients and/or the public were not involved in the design, or conduct, or reporting, or dissemination plans of this research.

**Patient consent for publication** Not required.

**Provenance and peer review** Not commissioned; externally peer reviewed.

**ORCID iD**
Viktoria Gastens http://orcid.org/0000-0002-6421-9867

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
