## [Reviewer comments · BMJ Open]

ARTICLE DETAILS

TITLE (PROVISIONAL)	Development and validation of a life expectancy estimator for multimorbid older adults: a cohort study protocol
AUTHORS	Gastens, Viktoria; Del Giovane, Cinzia; Anker, Daniela; Feller, Martin; Syrogiannouli, Lamprini; Schwab, Nathalie; Bauer, Douglas; Rodondi, Nicolas; Chiolero, Arnaud

VERSION 1 – REVIEW

REVIEWER	Aguado, Alba Consorti Sanitari Integral, CAP Sagrada Familia
REVIEW RETURNED	16-Feb-2021

GENERAL COMMENTS	The study protocol seems interesting and useful. An additional limitation to be mentioned might be that the patients were included at hospitalization and the model might not predict life expectancy so well if it is used in a different setting, such as primary care where preventive care is so common.
---

REVIEWER	Chudasama, Yogini University of Leicester College of Medicine Biological Sciences and Psychology, Diabetes Research Centre
REVIEW RETURNED	01-Mar-2021

GENERAL COMMENTS	From a statistical perspective, this study design has the following flaws which will highly affect the life expectancy predictions: 1. The small sample size of participants used to predict life expectancy (n=822). This will give an imprecise estimate when estimating remaining life expectancy and wide 95% confidence intervals.2. The follow-up time is very short, as only 3 years has been considered. The extrapolation on this is risky.3. Using the Cox proportional hazard instead of flexible parametric survival model. The flexible parametric survival model allows for greater flexibility to accurately capture data and extrapolate for future predictions.4. It is unclear which multimorbidity conditions are included in the definition of multimorbidity.5. Validation process is not clear.
--

VERSION 1 – AUTHOR RESPONSE

Reviewer #1

The study protocol seems interesting and useful.

Authors' response

Thank you.

An additional limitation to be mentioned might be that the patients were included at hospitalization and the model might not predict life expectancy so well if it is used in a different setting, such as primary care where preventive care is so common.

Authors' response

We have now included this limitation in the 'perspective' section (see page 8): "As the study participants were included at the time of hospitalization, they may not be representative of all patients with multimorbidity, and this could reduce the transportability of our findings to other populations."

Reviewer #2

From a statistical perspective, this study design has the following flaws which will highly affect the life expectancy predictions:

1. The small sample size of participants used to predict life expectancy (n=822). This will give an imprecise estimate when estimating remaining life expectancy and wide 95% confidence intervals.

Authors' response

We consider the sample size and related power adequate to address our research question. We have calculated the required sample size for conducting our multivariable prediction model utilizing the criteria proposed by Riley et al. To have a better clinical sense of the statistical power of our study, we have now added the following power calculation (page 7): "To have a better clinical sense of the statistical power of our study, we estimated the power to detect a between-group difference in LE that can be considered as important on a clinical point of view. If we stratify our sample in 3 groups of equal number of patients (N=274) with a relatively short, average, and long LE, the power to detect a statistically significant between-group difference of LE of 2 to 5 years is indicated in Table 3:

Table 3: Power to detect a statistically significant between-group difference in LE along various assumptions (N=274 per group; alpha = 0.05; two-sided). SD: standard deviation

Difference in LE (years) SD of LE (years) Power

2 4 >99%

2 8 83%

5 4 >99%

5 8 >99%

The relatively high power in all scenarios indicates that the sample size is adequate to address the research question."

2. The follow-up time is very short, as only 3 years has been considered. The extrapolation on this is risky.

Authors' response

We agree that it would have been better to have a longer follow-up for measuring long term mortality and LE. Nevertheless, it worth noticing that Lee et al. have developed an LE estimator from a mortality risk prognostic index and they used a 4-year mortality risk index (Lee et al., 2014). By comparison with the study by Lee et al, since we have a higher mortality, we have sufficient power to

conduct the planned analyses with sufficient confidence. In the 'perspective' section, we have added the following limitation (see page 8): “Further, we have a relatively short duration of follow-up.”

3. Using the Cox proportional hazard instead of flexible parametric survival model. The flexible parametric survival model allows for greater flexibility to accurately capture data and extrapolate for future predictions.

Authors’ response

We agree to apply the flexible parametric survival model and have revised the manuscript accordingly (see page 2 and 7): “The relationship between candidate predictors and outcome will be analysed using the flexible parametric survival model.”

4. It is unclear which multimorbidity conditions are included in the definition of multimorbidity.

Authors’ response

We defined multimorbidity as presence of 3 or more coexistent chronic medical conditions with an estimated duration of 6 months or more. The medical conditions were recorded as ICD-10 codes. The estimated duration was based on a clinical decision.

5. Validation process is not clear.

Authors’ response

We have specified the validation process in the 'methods' section (see page 7): “We will evaluate potential overfitting and optimism by internal validation with bootstrapping techniques. We will perform 500 bootstrap cycles in the original sample, resampling the same number of patients. In each bootstrap sample, we will derive a prediction model. We will calculate optimism as difference in performance measures between the original sample and the respective bootstrap sample. This will be repeated for all bootstrap samples to estimate the average optimism.”

References

Richard D Riley et al. “Calculating the sample size required for developing a clinical prediction model”. In: *BMJ* 368 (2020). doi: 10.1136/bmj.m441.

Sei J. Lee et al. “Individualizing Life Expectancy Estimates for Older Adults Using the Gompertz Law of Human Mortality”. In: *PLOS ONE* 9.9 (Sept. 2014), pp. 1–8. doi: 10.1371/journal.pone.0108540.

VERSION 2 – REVIEW

REVIEWER	Chudasama, Yogini University of Leicester College of Medicine Biological Sciences and Psychology, Diabetes Research Centre
REVIEW RETURNED	28-May-2021

GENERAL COMMENTS	This study still has major flaws which will indeed affect the life expectancy estimates. 1. The sample size is far too small. The authors have now presented a power calculation, but I am unsure how this has been calculated to find a difference between LE? It would be great if you could share the statistical codes? 2. The follow up is far too short to be making future extrapolations. Another study may have done this, but it does not make it correct.
---

	3. It is still unclear which multimorbidity conditions will be used? 4. I don't fully understand the validation process with a small sample size?
--	---

VERSION 2 – AUTHOR RESPONSE

Reviewer #2

Comments to the Author:

This study still has major flaws which will indeed affect the life expectancy estimates.

Authors' response

We are surprised by this statement and cannot agree. Indeed, the proposed study design has been carefully elaborated to fit our aim to provide an estimation of the life expectancy among multimorbid older adults based on the observed mortality risk. Please further note that our study protocol has been carefully peer-reviewed by 3 independent reviewers and by the funding agency (Swiss National Research foundation) committee. As part of the high-level reviewing process, we had to justify carefully the sample size and the follow-up duration. We have however refined our power calculation to avoid any misunderstanding.

1. The sample size is far too small.

The authors have now presented a power calculation, but I am unsure how this has been calculated to find a difference between LE? It would be great if you could share the statistical codes?

Authors' response

We calculated both sample size and power to address our research question and we found both adequate. Of note, key is the sample size calculation because we need to be sure that we can build a prediction model. The power calculation based on the potential between-group difference in estimated LE between groups of patients is provided to have an idea of what is possible with our sample size. We have added in the method section the following information (page 7): "We have calculated the required sample size for conducting our multivariable prediction model utilizing the criteria proposed by Riley et al. and implemented in the pmsampsize library for the R environment. For the 3-year mortality risk prognostic model, the minimum sample size required with 12 candidate predictor parameters, an expected outcome event rate of 0.1 per year, and an anticipated Cox-Snell R² 0.126 (C statistic of 0.82) is 799 with 20 events per predictor parameter. This is considerably more than the idiomatic 10 events per predictor parameter. Our sample size of 822 is therefore adequate for this project.

To have a better clinical sense of the statistical power of our study, we computed the power to detect a between-group difference in estimated LE that can be considered as important on a clinical point of view. If we stratify our sample in 3 groups of equal number of patients (N=274) with a relatively short, average, and long estimated LE, the power to detect a statistically significant between-group difference of LE of 1 to 3 years is indicated in Table 3:

Table 3: Power to detect a statistically significant between-group difference in LE along various assumptions (N=274 per group; alpha = 0.05; two-sided). SD: standard deviation

Difference in estimated LE (years) SD of LE (years) Power

1 4 83%

1 8 31%

2 4 >99%

2 8 83%

3 4 >99%

3 8 >99%

The relatively high power in most scenarios indicates that the sample size is adequate to address the research question.”

As required by the reviewer, we report below the codes used to calculate the sample size and power.

R code (version 4.1.0) for the sample size calculation:

```
library(pmsampsize)
```

```
# method: https://europepmc.org/article/med/30357870
```

```
# data, Prognostic Index for 4-Year Mortality in Older Adults:  
https://jamanetwork.com/journals/jama/fullarticle/202375
```

```
# validation cohort: C 0.82, E 1072, n 8009
```

```
# development cohort: C 0.84, E 1361, n 11701
```

```
# Calculating a sensible value for R2_CSapp from other reported information
```

```
# input C (C statistic) and E (number of reported events) and n (sample size)
```

```
C <- scan(n=1)
```

```
E <- scan(n=1)
```

```
n <- scan(n=1)
```

```
# calculate corresponding D statistic
```

```
D <- 5.50*(C - 0.5) + 10.26*(C - 0.5)^3
```

```

# calculate corresponding R2_D
R2_Dapp <- (pi/8*D^2)/(pi^2/6 + pi/8*D^2)

# calculate corresponding R2_OQuiley, by taking R2_Dapp as a proxy for R2_Royston
R2_OQuiley <- (-pi^2/6*R2_Dapp)/((1 - pi^2/6)*R2_Dapp - 1)

# calculate corresponding LR statistic
LR <- - E*log(1-R2_OQuiley)

# calculate R2_CSapp
R2_CSapp <- 1 - exp(-LR/n)

# input p (number of predictor parameters) and eventrate (event rate)
p <- scan(n=1)
eventrate <- scan(n=1)
years <- scan(n=1)
n_needed <-
pmsampsize(type="s",rsquared=R2_CSapp,parameters=p,rate=eventrate,timepoint=years,meanfup=
years)

#R code (version 4.1.0) for the power calculation:
power.t.test(n=274,delta=1,sd=4,sig.level=0.05,alternative="two.sided")
power.t.test(n=274,delta=1,sd=8,sig.level=0.05,alternative="two.sided")
power.t.test(n=274,delta=2,sd=4,sig.level=0.05,alternative="two.sided")
power.t.test(n=274,delta=2,sd=8,sig.level=0.05,alternative="two.sided")
power.t.test(n=274,delta=3,sd=4,sig.level=0.05,alternative="two.sided")
power.t.test(n=274,delta=3,sd=8,sig.level=0.05,alternative="two.sided")

```

2. The follow up is far too short to be making future extrapolations. Another study may have done this, but it does not make it correct.

Authors' response

It is not clear what is meant by “future extrapolations”. Our goal is to estimate life expectancy (LE) based on the information we observe on mortality after 3 years of follow-up, as done by Lee et al. among a younger population (older adults age 50 and over) followed-up over 4 years (Lee et al., 2014). We are conscious that the LE will be merely estimated based on the mortality risk assessed over 3 years of follow-up and that we do not have information on the cohort beyond this follow-up time. Key is however to have a valid 3 year mortality prognostic index, and this is what is expected with our study; LE is always an estimate, whatever the method used to compute it. One major strength of our study is to have data on older (70 years and over) multimorbid adults. Such a population is, to date, poorly investigated and often excluded from large scale trials. We agree that it would be better to have a longer follow-up for measuring long term mortality and to be more confident about LE. However, we believe that due to the high short-term mortality of this study population (up to 30% over 3 years), we have the possibility to make an estimation of the LE on over a reasonable follow-up duration.

3. It is still unclear which multimorbidity conditions will be used?

Authors' response

There are multiple ways to define multimorbidity (Ho et al., 2021). We have added the definition of multimorbidity to the manuscript (see page 4): “We defined multimorbidity as presence of 3 or more coexistent chronic medical conditions with an estimated duration of 6 months or more. The medical conditions were recorded as ICD-10 codes. The estimated duration was based on a clinical decision. The ten most frequent chronic medical conditions were in descending order: essential hypertension (I10), type 2 diabetes mellitus (E11), hypertensive heart disease (I11), chronic kidney disease stage 3 (N18.3), disorders of lipoprotein metabolism and other lipidaemias (E78), atrial fibrillation and flutter (I48), chronic ischaemic heart disease (I25), hyperplasia of prostate (N40), other postsurgical states (Z98), and obesity (E66).”

4. I don't fully understand the validation process with a small sample size?

Authors' response

We will perform an internal validation of the mortality prognostic index following the Prognosis Research Strategy (PROGRESS) framework (Steyerberg et al., 2013) and report it following the Transparent Reporting of a multivariable prediction model for Individual Prognosis Or Diagnosis (TRIPOD) statement (Collins et al., 2015). We have specified the internal validation process of the mortality prognostic index in the 'methods' section (see page 7): “We will use bootstrapping techniques to internally validate our mortality prognostic index. We will perform 500 bootstrap cycles from the original sample, sampling the same number of patients with replacement. In each bootstrap sample, we will derive a mortality prediction model and the relative risk score, as done in the original sample. We will also evaluate potential overfitting and optimism. We will calculate optimism as difference in performance measure (e.g. C-Statistics) between the original sample and the respective bootstrap sample. This will be repeated for all bootstrap samples to estimate the average optimism.”

Bootstrapping is a classical method for internal validation of prediction models (Moons et al., 2012). Among others, Moons et al. even especially recommend bootstrapping for models with a relatively small sample size in their highly esteemed review of methods for the development and internal validation of risk prediction models. As stated by Moons et al. “Bootstrapping is therefore the

preferred method for internal validation, certainly when the development sample is relatively small and/or a high number of candidate predictors is studied.”

References

Riley RD et al. “Calculating the sample size required for developing a clinical prediction model”. In: *BMJ* 368 (2020). doi: 10.1136/bmj.m441.

Lee SJ et al. “Individualizing Life Expectancy Estimates for Older Adults Using the Gompertz Law of Human Mortality”. In: *PLOS ONE* 9.9 (2014). doi: 10.1371/journal.pone.0108540.

Ho ISS et al. “Examining variation in the measurement of multimorbidity in research: a systematic review of 566 studies”. In: *The Lancet Public Health* (2021). doi: 10.1016/S2468-2667(21)00107-9.

Steyerberg EW et al. "Prognosis Research Strategy (PROGRESS) 3: prognostic model research," In: *PLoS medicine* 10 (2013). doi: 10.1371/journal.pmed.1001381.

Collins GS et al. "Transparent reporting of a multivariable prediction model for individual prognosis or diagnosis (TRIPOD): the TRIPOD statement," In: *BMJ* 350 (2015). doi: 10.1136/bmj.g7594.

Moons KG et al. “Risk prediction models: I. Development, internal validation, and assessing the incremental value of a new (bio)marker”. In: *Heart* 98 (2012). doi: 10.1136/heartjnl-2011-301246.